# Vestibular Compensation after Vestibular Dysfunction Induced by Arsanilic Acid in Mice

**DOI:** 10.3390/brainsci9110329

**Published:** 2019-11-18

**Authors:** Taeko Ito, Kouko Tatsumi, Yasumitsu Takimoto, Tadashi Nishimura, Takao Imai, Toshiaki Yamanaka, Noriaki Takeda, Akio Wanaka, Tadashi Kitahara

**Affiliations:** 1Department of Otolaryngology-Head and Neck Surgery, Nara Medical University, Kashihara 634-8522, Japan; t-nishim@naramed-u.ac.jp (T.N.); toshya@naramed-u.ac.jp (T.Y.); tkitahara@naramed-u.ac.jp (T.K.); 2Department of Anatomy and Neuroscience, Nara Medical University, Kashihara 634-8521, Japan; radha815@naramed-u.ac.jp (K.T.); akiow@naramed-u.ac.jp (A.W.); 3Department of Otolaryngology-Head and Neck Surgery, Osaka University Graduate School of Medicine, Suita 565-0871, Japan; yas.say.yas1.24@gmail.com (Y.T.); timai@ent.med.osaka-u.ac.jp (T.I.); 4Department of Otolaryngology, University of Tokushima School of Medicine, Tokushima 770-8503, Japan; takeda@tokushima-u.ac.jp

**Keywords:** mouse, labyrinthectomy, *p*-arsanilic acid, c-Fos, Arc, Zif268, vestibular nucleus, head deviation, nystagmus, vestibular compensation

## Abstract

When vestibular function is lost, vestibular compensation works for the reacquisition of body balance. For the study of vestibular dysfunction and vestibular compensation, surgical or chemical labyrinthectomy has been performed in various animal species. In the present study, we performed chemical labyrinthectomy using arsanilic acid in mice and investigated the time course of vestibular compensation through behavioral observations and histological studies. The surgical procedures required only paracentesis and storage of 50 µL of *p*-arsanilic acid sodium salt solution in the tympanic cavity for 5 min. From behavioral observations, vestibular functions were worst at 2 days and recovered by 7 days after surgery. Spontaneous nystagmus appeared at 1 day after surgery with arsanilic acid and disappeared by 2 days. Histological studies revealed specific damage to the vestibular endorgans. In the ipsilateral spinal vestibular nucleus, the medial vestibular nucleus, and the contralateral prepositus hypoglossal nucleus, a substantial number of c-Fos-immunoreactive cells appeared by 1 day after surgery with arsanilic acid, with a maximum increase in number by 2 days and complete disappearance by 7 days. Taken together, these findings indicate that chemical labyrinthectomy with arsanilic acid and the subsequent observation of vestibular compensation is a useful strategy for elucidation of the molecular mechanisms underlying vestibular pathophysiologies.

## 1. Introduction

The vestibular system contributes to posture and oculomotor control, perception, and the autonomic nervous system. Dysfunction of the vestibular system induces feelings of vertigo/dizziness, nausea, postural disorders, and ocular motor deficits. These functional deficits recover gradually and almost completely, without any vestibular peripheral regeneration, within a week in rats, 6 weeks in cats, and 3 months in humans; this recovery phenomenon is referred to as vestibular compensation [1,2].

Currently, unilateral vestibular neurectomy, unilateral surgical labyrinthectomy, and unilateral chemical labyrinthectomy are widely performed in animals such as rabbits, cats, and rats for the creation of animal models of vestibular dysfunction [3]. For a detailed investigation of molecular mechanisms, transgenic and gene-knockout mice are indispensable. However, the relatively small body size of mice precludes complex surgical procedures. Among the above-mentioned procedures, chemical labyrinthectomy is considered suitable for mice because it is simple and requires no special surgical technique. Although previous studies have observed behavioral changes after chemical labyrinthectomy in mice [4,5], few have described the process of vestibular compensation through objective observations in this species.

Kim et al. reported detailed procedures of chemical labyrinthectomy using arsanilic acid in rats [6]. They also described the process of vestibular compensation after chemical labyrinthectomy in rats and compared it with the process after surgical labyrinthectomy through assessments of nystagmus, head tilt, and distribution patterns for c-Fos-immunoreactive (IR) cells. Although nystagmus, head tilt, and c-Fos-IR cells were observed in cases of chemical labyrinthectomy as well as those of surgical labyrinthectomy, the time courses of compensation were quite different.

We performed a pilot study in mice using the same procedures described by Kim et al. [6], and preliminary experiments revealed some differences in the expression of nystagmus and head tilt after chemical labyrinthectomy with arsanilic acid. We then modified the procedure. In the present study, we used this modified procedure for unilateral labyrinthectomy (UL) with arsanilic acid and confirmed that the application of arsanilic acid in the tympanic cavity could reproduce mouse models of vestibular dysfunction, as judged by behavioral changes. In addition, we precisely recorded eye movements and spontaneous nystagmus during vestibular compensation using the same methods described by Imai et al. [7], despite difficulties in tracking the eye movements. We also observed the expression of c-Fos and other immediate early genes (IEGs) such as Arc and Zif268. We describe how these parameters collectively changed during vestibular compensation after chemical labyrinthectomy with arsanilic acid in mice.

## 2. Materials and Methods

### 2.1. Animals

Fifty-nine 8-week-old male C57BL/6N mice were purchased from Japan SLC and housed in standard cages under a 12-h light/dark cycle with temperature-controlled conditions. Food and water were available ad libitum. All procedures were approved by the Animal Care and Use Committee at Nara Medical University in accordance with the policies established in the NIH Guide for the Care and Use of Laboratory Animals. All efforts were made to minimize the numbers of animals used and their suffering.

### 2.2. Surgical Procedures

All surgical procedures were performed under deep anesthesia induced by isoflurane. Isoflurane inhalation anesthesia (3% for induction, 1.5% for maintenance; applied over the mouth and nose mixed with room air at a flow rate of 1 L/min) provided a convenient method for humane performance of short procedures. Alert behavior was observed less than 10 min after removal of the gas. This time frame allowed for behavioral observations shortly after UL. In addition, isoflurane has a minimal effect on the expression of IEGs [8]. To destroy the vestibular function of the animals, we applied arsanilic acid in the tympanic cavity (arsanilic acid group, *n* = 44; five for behavioral observations, three for physiological examinations and 24 for histological studies). Surgery was performed on the right ear using *p*-arsanilic acid sodium salt solution (0.4 M, pH 7.2; Tokyo Chemical Industry, Tokyo, Japan) in 0.1 M phosphate buffer (PB). A retroauricular incision was placed to expose the external ear canal, which was opened just anterior to the exit point of the facial nerve. Next, the tympanic membrane was caudally perforated up to the handle of the malleus using a 26-gauge needle [9]. After paracentesis, 50 µL of arsanilate solution was instilled into the tympanic cavity and removed after 5 min using a piece of experimental absorbent. Antibiotic cream (oxytetracycline hydrochloride) was applied to the subcutaneous tissue to prevent infection. The surgical wound was sutured, and the mice were allowed to recover in light. A vehicle group (*n* = 44; five for behavioral observations, three for physiological examinations and 36 for histological studies) received a transtympanic application of 50 µL PB (pH 7.1) via the same route.

### 2.3. Behavioral Observations

Five mice from the arsanilic acid group and five from the vehicle group were used for behavioral observations. The measurements were made just before and 1, 2, 3, 5, and 7 days after surgery. We used the clinical scoring system developed by Cassel et al. to assess unilateral vestibular syndrome [5].

### 2.4. Vestibular Signs (Open Field)

In the open field, we observed vestibular signs such as circling and muscle dystonia. Circling represents a stereotyped rotatory movement in circles around the hips of the animal, while muscle dystonia represents hypertonia on the side of the lesion. These behaviors were scored from 0 to 3 as follows: 0, no visible sign; 1, subtle presence of the sign; 2, clear evidence of the sign; and 3, the maximum expression of the sign.

### 2.5. Tail-Hanging and Landing Test

For the tail-hanging and landing test, we held the mice by their tails and lifted them vertically over a height of approximately 50 cm. This test normally induces forelimb extension as the animals reach the ground, and unilateral vestibular deficit causes difficulty during the landing process. The responses of the animals while landing were scored from 0 (perfect preparation of the two front paws before reaching the ground) to 3 (no preparation for landing). The landing process was accompanied by axial rotation of the body, which was also scored from 0 (no rotation) to 3 (continuous twisting). Finally, the intensity of symptom reactivation after landing was scored from 0 (no sign) to 3 (maximum expression/accentuation of circling, tumbling, muscle dystonia, bobbing, and/or head tilt).

### 2.6. Head Deviation

Head deviation, defined as the angle between the horizontal plane and a line passing through the center of the animal’s head in the coronal plane (Figure 1H), was measured once a day.

### 2.7. Nystagmus

Nystagmus was observed as a vestibular sign. Three days before UL, the mice were anesthetized with isoflurane as described above. We placed a small incision on the head skin and fixed a small metal plate with a screw hole in the center of the skull using dental cement (Sun Medical, Moriyama, Japan). After surgery, eye movements were analyzed as previously described [7]. Briefly, each mouse was placed in a plastic cylinder with a metal bar on a square. The metal plate on the mouse’s head was screwed on to the bar on the tube under inhalation anesthesia with 1.5% isoflurane. Ten minutes after awakening, eye movements were recorded by a digital video camera (STC-CL338A, Omron Sentech, Kyoto, Japan). The custom-built metal plate, cylinder tube, camera holder, and all other custom equipment were manufactured by Bio-Medica (Kyoto, Japan). JPEG digital images of eye movements were acquired using StreamPix software (ARGO CORPORATION, Osaka, Japan). The software for analyzing eye movement was written in Visual C++ packaged in Visual Studio 2013 (Microsoft, Edmond, WA, USA). The right and left edge of the pupil were detected and then its center was decided. For the observation of eye movements, we tracked the center of the pupil. Nystagmus was recorded every 6 h.

### 2.8. Histology

#### 2.8.1. Temporal Bone Histology

After fixative perfusion at 1, 2, 7, or 30 days after surgery, pieces of temporal bone tissues were resected and dehydrated for 30 min at each step of an alcohol gradient: once at 30%, 50%, 70%, 85%, and 95%, and twice at 100%. Clearing was performed twice using xylol for 1 h. The specimens were subsequently immersed twice in liquid paraffin (1 h at 42–46 °C), blocked using solid paraffin in a paraffin cast, and incubated at 46–52 °C for 1 day. Deparaffinization was performed for 5 min at each step of an alcohol gradient: once at 30%, 50%, 70%, 85%, and 95% and twice at 100%. Subsequently, the specimens were rinsed twice with distilled water for 5 min. The tissues were cut into 4-μm-thick sections using a Microm HM340E (Thermo Fisher Scientific, Waltham, MA, USA), and cross-sections were stained with hematoxylin–eosin. Histological observation and image acquisition were achieved with an Olympus BX51 microscope (Olympus, Kyoto, Japan). Slices were also stained with Fluoro-Jade C (Fluoro-Jade C Ready-to-Dilute Staining Kit, Biosensis, Thebarton, Australia) to check cellular degeneration according to the manufacturer’s protocol.

#### 2.8.2. Silver Staining

After transcardial perfusions with 4% paraformaldehyde at 1, 2, 7, and 30 days after surgery, the brains of mice were dissected with their skulls and post-fixed in the same solution for 2 days at 4 °C. The brains and vestibular ganglions were then removed from the skulls. We performed post-fixation before collecting vestibular ganglions so as not to damage the cells of vestibular ganglions. Next, the vestibular ganglions were preserved in the same solution for a further 10 days. After fixation, they were placed in 30% sucrose solution in phosphate-buffered saline (PBS) until they sank. In the present study, we employed an amino-cupric silver staining technique [10] to detect degenerating neurons. FD NeuroSilver Kit II (FD NeuroTechnologies, Columbia, MD, USA) enabled us to perform the amino-cupric silver staining. According to the instruction of the kit, sections with a thickness of 80 µm were cut on a cryostat microtome (Leica CM 1850, Leica Biosystems, Tokyo, Japan) and stained. We checked the validity of the staining procedures using a slice of positive control tissue provided by the company (FD NeuroTechnologies, Columbia, MD, USA).

#### 2.8.3. c-Fos/Arc/Zif268 Immunohistochemistry

Mice were transcardially perfused with 4% paraformaldehyde and glutaraldehyde in 0.1 M PB at 1, 2, 3, 5, or 7 days after surgery. After fixative perfusion, the brainstems were removed and post-fixed in the same solution for 20 h at 4 °C. They were then snap-frozen after placement in 30% sucrose solution in PBS until they sank. Next, frozen sections with a thickness of 30 µm were cut using a cryostat microtome (Leica CM 1850, Leica Biosystems), and immunoperoxidase procedures were performed as previously described [11]. Briefly, endogenous peroxidase activity was blocked by incubation for 30 min in 0.1% H_2_O_2_ in PBS with 0.3% Triton X-100 (PBST). A nonspecific blocking procedure was performed with 5% normal goat serum (Vector Laboratories, Burlingame, CA, USA) in PBST for 30 min before the application of primary antibodies in different dilutions: anti-c-Fos (1:20,000, rabbit polyclonal, sc-52, Santa Cruz Biotechnology, Dallas, TX, USA), anti-Arc (1:20,000, rabbit polyclonal, Synaptic Systems, #156 003, Germany), and anti-Zif268 (1:15,000, rabbit polyclonal, Santa Cruz Biotechnology, sc-110). After incubation for 48 h, the sections were washed three times with PBST, followed by incubation (for 22 h at 4 °C) with goat anti-rabbit IgG combining amino acid polymers and peroxidase (Histofine simple stain PO Kit, Niterie Bioscience, Japan) diluted with PBST. The peroxidase color reaction was performed in diaminobenzidine tetrahydrochloride (DAB) solution (DAB Substrate Kit, Vector Laboratories, Burlingame, CA, USA).

For immunofluorescence double-labeling, frozen sections with a thickness of 30 µm were prepared as described above. After blocking with 5% normal donkey serum (NDS, Sigma Aldrich, Japan) in PBST for 30 min, the sections were incubated for 48 h with a cocktail of primary antibodies: anti-c-Fos (1:2000, guinea pig polyclonal, Synaptic Systems, #226 004) and anti-Arc (1:6000, rabbit polyclonal, Synaptic Systems, #156 003) or anti-c-Fos and anti-Zif 268 (1:6000, rabbit polyclonal, Santa Cruz Biotechnology, sc-110). Subsequently, they were washed three times with PBST, followed by incubation for 1 h with a cocktail of secondary antibodies: donkey Alexa 594-conjugated anti-guinea pig IgG and Alexa 488-conjugated anti-rabbit IgG (1:1000, Jackson Immunoresearch Laboratories, West Grove, PA, USA). Blue-fluorescent Nissl stain (1:200, Neurotrace 435/455, Thermo Fisher Scientific) was used for counterstaining.

All fluorescence images were obtained as z-stack projections (*x*-axis by *y*-axis, 1024 × 1024 pixels) using a confocal laser scanning microscope (C2-NiE, Nikon, Tokyo, Japan). We obtained z-stack multichannel images at 1-μm intervals using a 60× water-immersion objective lens (NA 1.45, UpanApo, Nikon) at an optical zoom of 1.5. The images generated orthogonal views (x–z and y–z planes; Appendix A) and were processed by Nikon Nis-Elements software version 4.11 (Nikon, Japan).

### 2.9. Cell Counting in the Spinal Vestibular Nucleus (SpVN), Medial Vestibular Nucleus (MVN), and Prepositus Hypoglossal Nucleus (PrHN)

For c-Fos-IR, Arc-IR, and Zif268-IR cell detection after surgery, 30-µm-thick brainstem sections were examined under a bright-field microscope (BX51, Olympus) with 40× and 100× objective lens. Only cells exhibiting a significant intensity of the DAB reaction product in their nuclei were counted as c-Fos-IR/Arc-IR/Zif268-IR cells. The DAB reaction products in the nuclei in SpVN, MVN, and PrHN from the vehicle group were adopted as the background for IEG-IR cells. We counted the number of IEG-IR cells in the ipsilateral and contralateral SpVN/MVN/PrHN through the rostral (Bregma: −5.52 mm) and caudal parts (Bregma: −7.08 mm), in accordance with a brain atlas [12]. To prevent duplicate counting, we adopted an edge effects unbiased cell counting method [13].

### 2.10. Statistical Analysis

One-way analysis of variance (ANOVA) was used for multiple comparisons, followed by pairwise comparisons with Tukey’s post hoc tests or paired Student’s t-tests for two-group comparisons. A *p*-value of <0.05 was considered statistically significant. All statistical analyses were performed using SPSS version 25.0 (IBM Corp., Armonk, NY, USA).

## 3. Results

### 3.1. Behavioral Observations

#### 3.1.1. Vestibular Signs (Open Field)

There was significant weight loss in the arsanilic acid group at 2 days after surgery (*p* < 0.05, paired t-test), although the body weight was restored by 5 days (Figure 1A). There was no weight change in the vehicle group. We observed circling and muscle dystonia as spontaneous vestibular signs in the open field in the arsanilic acid group. The circling score significantly increased at 2 days and gradually decreased from 2 to 7 days after surgery (maximum mean score, 2.8 ± 0.4; Figure 1B). The score for muscle dystonia showed a similar trend (maximum score, 2.8 ± 0.4; Figure 1C). Vestibular signs were not observed in the vehicle group.

#### 3.1.2. Tail-Hanging and Landing Test

Using the tail-hanging and landing test, we observed the quality of landing, position of the mouse body during up/down movements, and intensity of symptom reactivation after landing. The maximum landing score was observed at 2 days after surgery (2.4 ± 0.5; Figure 1D), following which it gradually decreased to normal by 7 days. The scores for axial rotation (Figure 1E) and symptom reactivation after landing (Figure 1F) also peaked at 2 days after surgery and returned to normal by 7 days (axial rotation, 2.6 ± 0.5; symptom reactivation, 2.8 ± 0.4). In contrast, the vehicle group did not show any vestibular signs during the tail-hanging and landing test (Figure 1B–G).

#### 3.1.3. Head Deviation

We compared the head deviation with that 1 day later. In the arsanilic acid group, the angle of head deviation began to decrease from 1 day after surgery, with the minimum value observed at 2 days. Although the angle changed significantly from 1 to 3 days after surgery, it remained unchanged on day 4. Thereafter, gradual recovery was observed, although the original level was not achieved even at 7 days (Figure 1I).

#### 3.1.4. Nystagmus

In the arsanilic acid group, spontaneous nystagmus was clearly observed (Figure 1J). The frequency of nystagmus was 22.0 beats/15 s at 24 h after surgery, and the maximum frequency (38.7 beats/15 s) was observed at 30 h. Subsequently, the frequency gradually decreased, and nystagmus disappeared by 48 h after surgery (Figure 1J,K). None of the animals in the vehicle group showed nystagmus (Figure 1K).

### 3.2. Temporal Bone Histology

Figure 2 shows hematoxylin–eosin-stained images of the temporal bone from mice in the arsanilic acid and vehicle groups. In the vehicle group, the sensory epithelia of the utricles, saccules, and crista ampullaris in the semi-circular canals did not show any visible damage. In the arsanilic acid group, however, these sensory epithelia showed a number of vacuoles, which were observed as early as 1 day after surgery and gradually increased in size by 7 days. They were seen below the sensory cell layers, near the synaptic contacts, and between the sensory cells and the neuronal processes arising from the vestibular ganglion neurons. The vacuoles were also seen in the supporting cell layer. Hair cells were no longer visible in the utricles and saccules at 2 days and 7 days after surgery in the arsanilic acid group. In the utricles and saccles, the disarrangement and loss of otoconia could be observed. In the crista ampullaris in the semi-circular canals, the epithelia displayed 37.5% hair cell loss at 7 days after surgery. Swelling of the calyx and bouton nerve terminals was observed throughout the utricle and crista ampullaris. Figure 3 shows images of a vestibular ganglion stained with hematoxylin–eosin, Fluoro-Jade C, and silver. Unlike the vestibular sensory organs, the vestibular ganglions from both groups showed no significant changes on hematoxylin–eosin staining, with intact cell bodies showing no swelling. Consistent with these findings, Fluoro-Jade C and silver staining showed only a few stained cells in specimens from both groups (Figure 3 and Appendix A). There was no significant damage to the hair cells of the cochlea and stria vascularis (Appendix A). We also evaluate the hearing function by auditory brainstem responses (ABRs). The thresholds of ABRs were elevated by 30–40 dB in both arsanilic acid and vehicle group, compared with intact group (Appendix A). The elevation of the thresholds showed only the effects of paracentesis and not the arsanilic acid.

#### c-Fos/Arc/Zif268 Expression in the Vestibular Nuclei

In the arsanilic acid group, c-Fos-IR cells were clearly and predominantly observed in the ipsilateral SpVN and MVN and in the contralateral PrHN after surgery (Figure 4D). There were few arsanilic acid-induced c-Fos-IR cells in any other central vestibular-related regions (Figure 4D). These findings were consistent with those of a previous study about c-Fos expression during vestibular compensation after UL in rats [14]. Figure 4B,C show changes in the number of c-Fos-IR cells in the ipsilateral and contralateral SpVN/MVN/PrHN. There were few c-Fos-IR cells in SpVN/MVN/PrHN in the vehicle group (Figure 4C). In the arsanilic acid group, the ipsilateral SpVN/MVN and contralateral PrHN showed a substantial number of c-Fos-IR cells at 1 day after surgery, with a maximum increase in number at 2 days after surgery and a gradual decrease to zero by 7 days (Figure 4B). The expression of Arc-IR cells was almost the same as that of c-Fos-IR cells, although the number was much smaller (Appendix A). Zif268-IR cells were found on both the ipsilateral and contralateral sides (Appendix A), similar to findings in previous reports [15,16], although the cell number on the ipsilateral side was significantly larger than that on the contralateral side (Appendix A). Some double-staining c-Fos-IR/Arc-IR and c-Fos-IR/Zif268-IR cells were observed in the ipsilateral MVN (Appendix A).

## 4. Discussion

The present study described the characteristics of vestibular compensation after chemical labyrinthectomy using arsanilic acid in mice. Because chemical labyrinthectomy with arsanilic acid needs only paracentesis, it is much simpler than surgical labyrinthectomy. Such a simple surgery is desirable for experiments in mice, which have very small bodies but are useful for molecular biological studies. In the present study, we used a retroauricular incision to reveal the tympanic membrane. In a pilot study, we tried the transcanal approach for observation of the tympanic membrane and paracentesis. However, vestibular signs such as head deviation and nystagmus differed among animals when this protocol was used (data not shown), probably because of differences in the amount and duration of storage of arsanilic acid in the tympanic cavities. The retroauricular approach in the present study provided a larger field of view and enabled direct delivery of arsanilic acid to the tympanic cavity and precise removal after a fixed period of time. Moreover, all mice showed similar vestibular signs. Therefore, the retroauricular approach is suitable for reproducible labyrinthectomy in mice.

Kim et al. reported that surgical and chemical labyrinthectomies led to quite different behavioral profiles during the processes of vestibular compensation in rats [6]. Surgical labyrinthectomy induced maximal effects on spontaneous nystagmus and head deviation immediately after UL in rats, whereas chemical labyrinthectomy using arsanilic acid caused a gradual increase in the frequency of spontaneous nystagmus and the degree of head deviation. In the present study, mice displayed spontaneous nystagmus at 24 h after surgery with arsanilic acid, a response similar to that observed for rats. The drug delivery system for arsanilic acid may have influenced these outcomes; *p*-arsanilic acid sodium salt solution in the tympanic cavity can diffuse to the vestibular epithelia through the oval or round window and, to a lesser degree, extend via blood vessels or lymphatics and show interscalar exchange, as described for application of a fluorescence-labeled endotoxin by Takumida et al. [17]. Vignaux et al. reported specific effects of arsanilic acid on the eighth nerve of rats [18]. The intratympanic application of arsanilic acid in rats did not lead to the degeneration of the eighth nerve within the brainstem to vestibular nucleus at 3 or 7 days after surgery. In the present study, we obtained the same results as Vignaux et al.: there was no degeneration of the eighth nerve and also no abnormally degenerated neurons in any other parts of brains in mice at 2 or 7 days after surgery (data not shown). These findings, together with the present histological findings (Figure 2 and Figure 3) of arsanilate-induced damage in the vestibular sensory organs without involvement of the vestibular ganglia, suggest the feasibility of this delivery route without diffusion into the sheath of the eighth cranial nerve. Our behavioral observations (Figure 1) and histopathological analyses (Figure 4) also suggest that arsanilic acid gradually spreads to the vestibular sensory organs, with no vestibular signs before 24 h. Our histropathological findings (Figure 2 and Figure 3) and ABRs (Appendix A) showed the arsanilic acid did not affect cochlear tissues and auditory activities, but were restricted within the vestibules. The characteristics can be used for the treatment of Meniere’s disease. For the treatment of Meniere’s disease, gentamicin, an ototoxic drug, can be applied into the tympanic cavity. In order to suppress the function of the vestibules and decrease the frequency of vertigo, gentamicin must be applied into tympanic cavities many times and may cause the hearing loss. On the other hands, the results of the present study showed one-time intratympanic application of arsanilic acid can destroy only vestibules without negative effects on the other organs. From these findings, intratympanic application of arsanilic acid could be a treatment of Meniere’s disease. In this case, however, the arsanilic acid, a toxin, may cause unexpected side effects when it leaks to systemic circulation.

In the present study, c-Fos-IR cells appeared bilaterally in the vestibular nucleus by 1 day after surgery with arsanilic acid and disappeared by 7 days, with the maximum increase at 2 days after surgery, as previously reported [14,19,20,21]. These c-Fos-IR cells were predominantly found in the ipsilateral SpVN and MVN and contralateral PrHN [14,20,21]. Unlike the findings of Cirelli et al., we found significant expression only in the ipsilateral SpVN and MVN and contralateral PrHN [19]. This discrepancy can be attributed to differences in the type of labyrinthectomy (surgical or chemical) and the animal species (rats or mice) between the two studies. Because few papers have described changes in the expressions of IEGs other than c-Fos during the process of vestibular compensation, Arc and Zif268 were also examined in the present study. They predominantly localized in the ipsilateral SpVN and MVN and contralateral PrHN, having a distribution pattern similar to that shown by UL-induced c-Fos-IR cells. These findings strongly suggest that all IEG-IR cells are candidate indicators for the process of vestibular compensation after UL. As illustrated in Figs. 4B and S3B, the three IEGs (c-Fos, Arc, and Zif268) showed somewhat different profiles. IEGs are induced immediately after neuronal activation and participate in diverse functions. Arc, one of the effector IEGs, is directly involved in cellular changes at locations such as the cytoskeleton or receptors [22], while Zif268, a regulatory transcription factor, is responsible for inducing the transcription of late-response genes [23]. Most IEGs are located and synthesized in the cell soma, although the transcripts of Arc are transported to the dendrites and translated there in accordance with n-methyl-d-aspartate (NMDA) receptor activation [24,25]. Temporal differences in the expression patterns of these IEGs suggest that c-Fos, Arc, and Zif268 take part in different stages at different sites in the process of vestibular compensation.

The profiles of behavioral deficits observed in our arsanilic acid group (Figure 1I,K) were comparable with those of IEGs in the vestibular nucleus after UL (Figs. 4B and S4B). PrHN receives input fibers from the bilateral frontal eye fields, optic fascicular nuclei, and contralateral MVN, and sends its projection fibers to all bilateral motor nervous nuclei governing the extraocular muscles [12]. Several previous studies have indicated that c-Fos could be a good marker of UL-induced activation of rebalancing neurons in MVN and PrHN [14,21,26,27]. Taking these data together with the present findings, we suggest that c-Fos-IR cells in the ipsilateral MVN and contralateral PrHN are activated through UL-induced disinhibition mechanisms in the vestibular commissure and/or vestibulo-cerebellum and have positive effects on UL-induced vestibulo-ocular deficits. Recovery from vestibulospinal deficits was delayed relative to recovery from vestibule-ocular deficits, in accordance with the relatively slow decrease in the expression of UL-induced Arc-IR cells. SpVN and MVN are strongly associated with the vestibulospinal reflex and vestibulo–ocular reflex (VOR), respectively. Therefore, Arc-IR cells in the ipsilateral SpVN after UL may have some influence on UL-induced vestibulospinal deficits. Tighilet et al. reported that downregulation of histamine H3 receptors in the medial vestibular nucleus after UL could facilitate GABA release from cerebellar inputs and inhibitory interneurons that make synaptic contacts with second-order neurons; alternatively, it could facilitate glutamate release from the terminals of second-order MVN neurons that synapse on inhibitory interneurons in the contralateral MVN [28]. Modulation of GABAergic and glutamatergic neurons by H3 receptors should restore the balance between the vestibular nuclei on both sides. This is in agreement with the findings of Kitahara et al., who showed that NMDA receptors mediated the inhibitory neural circuits and played important roles in the restoration of balance between intervestibular nuclear activities [26]. As noted above, Link et al. reported that Arc transcripts are transported to the dendrites and translated there in accordance with NMDA receptor activation [24,25]. All these findings suggest that Arc, a prominent IEG, is a good indicator of the mechanisms of vestibular compensation.

Vertigo and dizziness are common symptoms and among the most frequent reasons for medical consultation. In a meta-analysis, a lifetime prevalence of vertigo with a vestibular disorder was seen in 3% to 10% of the total population [29]. The vestibular symptoms after a unilateral vestibular disorder comprise both static symptoms, which are observed at rest, and dynamic symptoms, which are observed on movement. Static symptoms are usually compensated perfectly in patients with a unilateral vestibular disorder [30,31], whereas dynamic deficits often remain poorly compensated and persist over a longer period [30,31]. In such cases, a drop in the gain and phase shifts of VOR and a decrease in the time constant of VOR are observed. Clinical studies have shown that vestibular rehabilitation is effective for the recovery of VOR and the promotion of dynamic vestibular compensation [32,33]. However, the mechanisms underlying such improvements remain unclear, and it is unknown why the same rehabilitation method leads to recovery in some patients but shows no effects in others [33]. For a deep understanding of the effects of vestibular rehabilitation, the entire process of vestibular compensation, from the beginning of static vestibular compensation to the end of dynamic vestibular compensation, needs to be studied in animal models, particularly mice. The start of static compensation can be observed by assessing c-Fos-IR cells [14,20,34]. The period of complete static compensation and incomplete dynamic compensation after UL can recognized by the expression of c-Fos-IR cells using MK801, an NMDA receptor antagonist. During this time, inhibitory neuronal circuits from the side of the lesion to the contralateral side may be switched on to achieve rebalance between intervestibular neuronal activities [21,26,27]. Moreover, MK801 may induce the expression of c-Fos-IR cells again, predominantly in the contralateral MVN/ipsilateral PrHN, along with behavioral deficits of head deviation and spontaneous nystagmus; this is referred to as vestibular decompensation [21,26,27]. The gain of VOR can be an indicator of the promotion of dynamic vestibular compensation, with complete recovery indicating complete dynamic vestibular compensation. In the present study, we confirmed the expression of c-Fos-IR cells during vestibular compensation, particularly static vestibular compensation, after chemical labyrinthectomy with arsanilic acid in mice. Using MK801, we should be able to reveal the condition of the vestibular nucleus during the period of complete static vestibular compensation and incomplete dynamic vestibular compensation. In the present study, we could observe eye movements and nystagmus after chemical labyrinthectomy in mice. This facilitates the observation of VOR and visualization of the process of dynamic vestibular compensation. Taking our findings together, we believe that the entire process of vestibular compensation after UL in mice can be observed.

Recently, molecular biological strategies have been developed using gene manipulation with various types of knockout and/or knock-in mice for genes in the central and/or peripheral nervous system. These strategies have also been applied in vestibular compensation experiments [35,36]. However, mice are small animals, and great surgical skill is required to perform labyrinthectomy and collect the inner ear tissues. This has limited the progress of scientific findings in this field. The procedures demonstrated in the present study are not difficult and do not require special surgical training. We believe that their application will stimulate further progress in elucidation of the molecular mechanisms of vestibular compensation.

## Figures and Tables

**Figure 1 brainsci-09-00329-f001:**
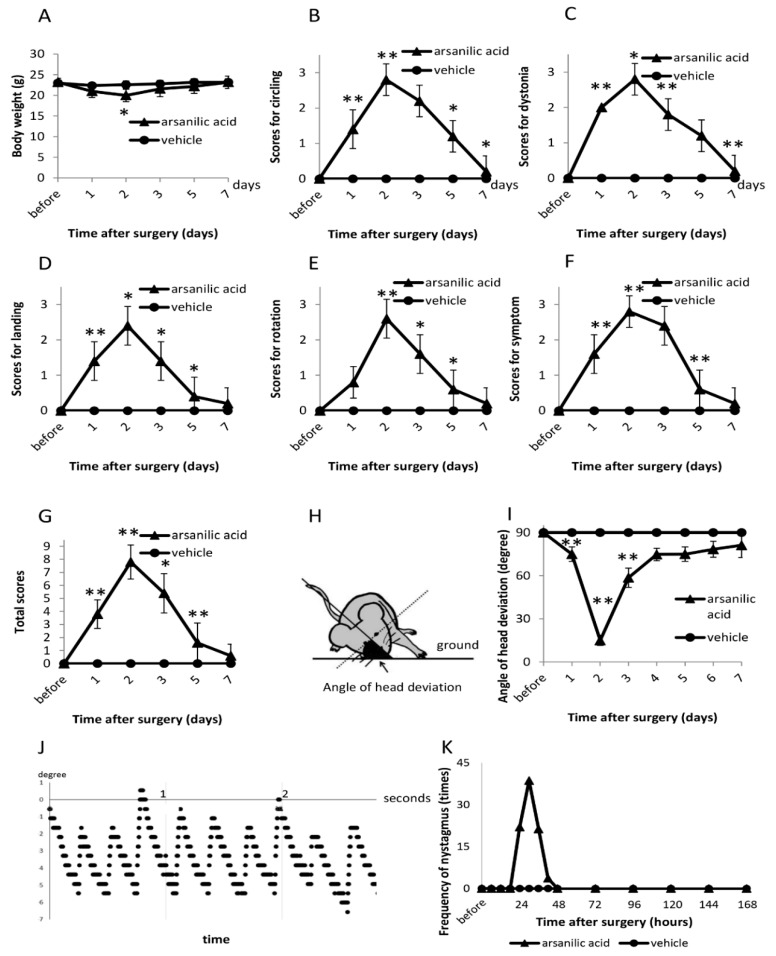
Evaluation of vestibular signs after surgery in the arsanilic acid (unilateral labyrinthectomy with arsanilic acid) and vehicle groups (unilateral labyrinthectomy with phosphate buffer). (**A**) Changes in weight after surgery. (**B**) Scores for circling. (**C**) Scores for muscle dystonia. (**D**) Scores for landing in the tail-hanging and landing test. (**E**) Scores for axial rotation in the tail-hanging and landing test. (**F**) Scores for symptom reactivation after the tail-hanging and landing test. (**G**) Total scores for the tail-hanging and landing test. (**H**) Measurement of the angle of head deviation. (**I**) Angle of head deviation after surgery. (**J**) Typical nystagmus at 30 h after surgery. (**K**) Frequency of nystagmus after surgery. In the arsanilic acid group, significant weight loss is seen at 2–3 days after surgery, with gradual recovery by 5 days (**A**). In the open field, vestibular signs are strongest at 2 days after surgery, followed by gradual recovery to normal by 7 days (**B**,**C**). In the tail-hanging and landing test, the arsanilic acid group shows maximum scores at 2 days after surgery, with return to normal by 7 days (**D**–**G**). The angle of head deviation is the smallest at 2 days after surgery, with gradual but not complete recovery by 7 days (**I**). Nystagmus appears at 24 h after surgery and disappears by 48 h (**K**). * *p* < 0.05, ** *p* < 0.01, paired t-test (vs. the previous time point of observation).

**Figure 2 brainsci-09-00329-f002:**
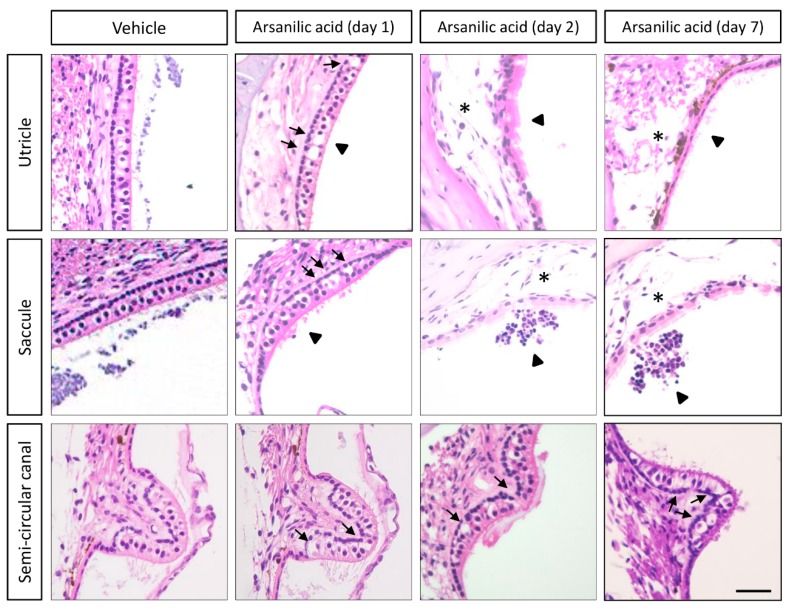
Histological observations (hematoxylin–eosin staining) after unilateral labyrinthectomy with arsanilic acid or phosphate buffer (vehicle) in mice. The vehicle group shows negligible damage in the utricles, saccules, and semi-circular canals. In the arsanilic acid group, vacuoles can be seen in the utricles and saccules at 1 day after surgery (arrows). The vacuoles gradually increase in size, and the macular structures are destroyed by 2 days after surgery (asterisk). The vacuoles are also observed in the semi-circular canals at 1, 2, and 7 days after surgery (arrows). In the utricles and saccules, the disarrangement and loss of otoconia could be observed (arrowheads). Scale bar, 20 μm.

**Figure 3 brainsci-09-00329-f003:**
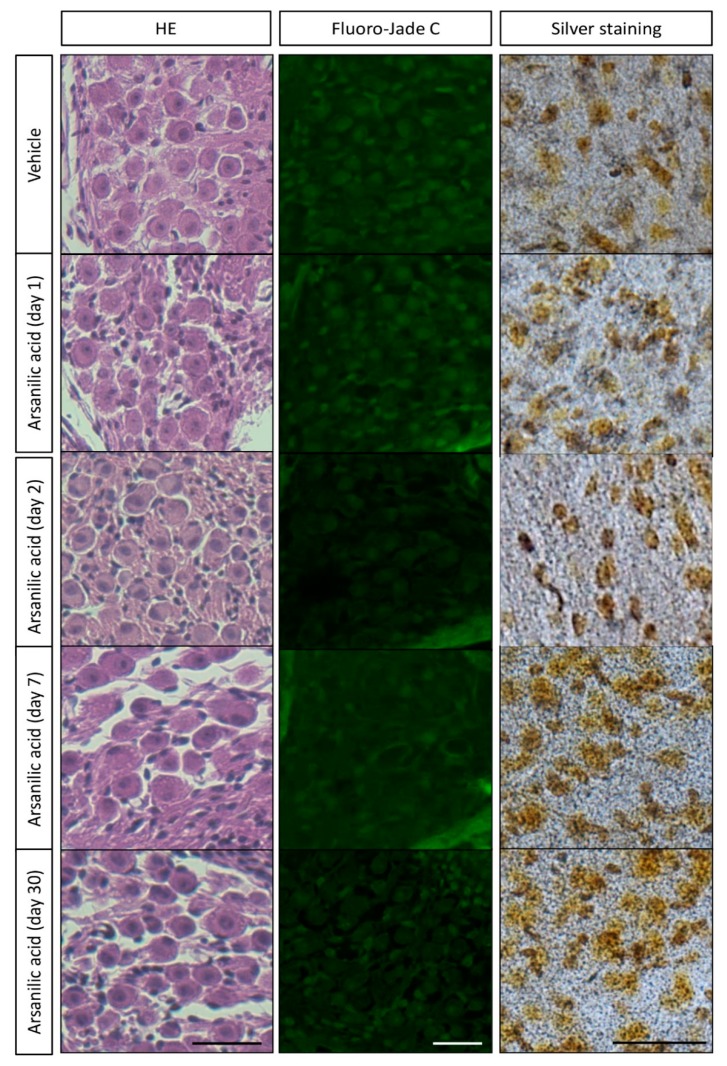
Observation of cells in mouse vestibular ganglions after staining with hematoxylin–eosin, Fluoro-Jade C, and silver. Paraffin sections of temporal bone were stained with hematoxylin–eosin (HE) and Fluoro-Jade C to check cellular integrity and degeneration. We further employed an amino-cupric silver staining method that is very sensitive in detection of cellular degeneration. For the silver staining, we used both frozen sections and vibratome sections of the vestibular ganglions. Because the frozen sections were of uniform thickness and gave us clearer images, representative pictures of frozen sections are shown here. There is no difference in the amount of structural damage observed with hematoxylin–eosin staining, or in the number of cells stained by Fluoro-Jade C and silver, between the arsanilic acid (unilateral labyrinthectomy with arsanilic acid) and vehicle (unilateral labyrinthectomy with phosphate buffer) groups. Scale bar, 20 μm.

**Figure 4 brainsci-09-00329-f004:**
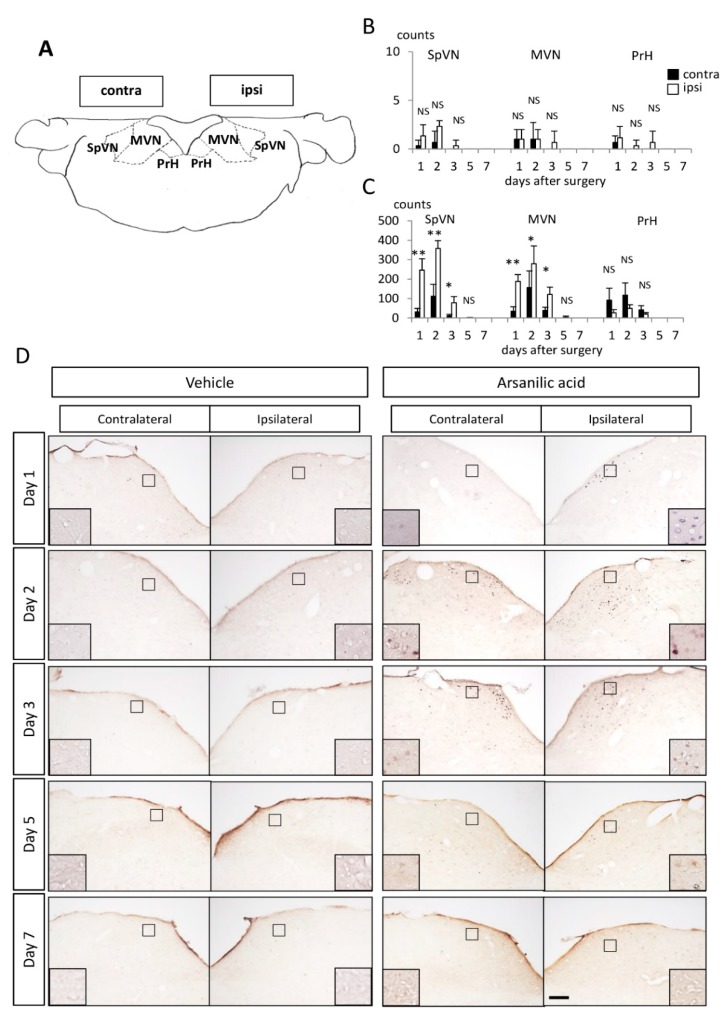
Expression of c-Fos-IR cells in the ipsilateral and contralateral spinal vestibular nucleus (SpVN), medial vestibular nucleus (MVN), and prepositus hypoglossal nucleus (PrHN). (**A**) Vestibular nucleus. (**B**) Number of c-Fos-IR cells in the vestibular nucleus in the vehicle (unilateral labyrinthectomy with phosphate buffer) group. (**C**) Number of c-Fos-IR cells in the vestibular nucleus in the arsanilic acid (unilateral labyrinthectomy with arsanilic acid) group. (**D**) Expression of c-Fos-IR cells in the vestibular nucleus. There are few c-Fos-IR cells in SpVN/MVN/PrHN in the vehicle group (**B**,**D**). In the ipsilateral and contralateral SpVN/MVN/PrHN in the arsanilic acid group, a substantial number of c-Fos-IR cells can be seen on day 1 after surgery, with a maximum increase in number at 2 days and a gradual decrease to zero by 7 days (**C**,**D**). * *p* < 0.05, ** *p* < 0.01. NS, not significant; one-way analysis of variance (ipsilateral vs. contralateral); c-Fos-IR cells were predominantly found in the ipsilateral SpVN and MVN and the contralateral PrHN. The box inserts show representative c-Fos-IR cells. Scale bar, 50 μm at lower magnification.

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
