# Peer review of "Vestibular Compensation after Vestibular Dysfunction Induced by Arsanilic Acid in Mice"

_brainsci, 2019, doi:10.3390/brainsci9110329_

Round 1

Reviewer 1 Report

I congratulate the authors for their novel methods which introduce a useful model for the ongoing study of acute vestibular impairment, recovery and treatment in an animal model.

I do not have any additional comments for the review.

I hope to see this article in press.

Author Response

We thank the reviewer for the warm comments. We are glad if the results of the present study could contribute to the development in the field of study of vestibular disorders.

Once again, we sincerely appreciate the reviewer’s comments.

Reviewer 2 Report

This is an interesting study presenting a mouse model of chemical labyrinthectomy using arsanilic acid. The study is well-written and present behavioural, functional and histopathological data.

Minor revisions

Figure 1 contains 11 subfigures and some of them should be improving in the labeling for a better understanding. So B panel should be marked with circling or C panel dystonia, D landing test and so on.

Questions

1) Did the authors perform any hearing test in mice? If not they should state that in the limitations of the study.

2) Did you observed any changes in the otolithic membranes such as loss of otoconia?

3) In the discussion, the authors should mention the potential use of arsanilic acid for intratimpanic application in Meniere disease. This should be compared with the current approach of intratympanic gentamicin in intractable Meniere disease.

Author Response

Figure 1

We thank the reviewer for the suggestion. We modified the labeling according to the suggestion.

Questions

1)

We thank the reviewer for the comment. We have performed the ABR and added the data in the Supplementary Fig. 3 with legends. The thresholds were elevated by 30 – 40 dB in both arsanilic acid and vehicle group, compared with intact group. We thought paracentesis could cause the slight elevation and not the arsanilic acid. We described these in result section (Page 9, 267-271).

We also added a description “The arsanilic acid did not affect the cochlear tissue and auditory activities” in the Discussion Section (Page 14, lines 354-355).

2)

Yes, we did observe the disarrangement or loss of otoconia in the utricles and saccules (Fig. 2 arrowheads). We added this in the result section and legend of Fig.2.

3)

We thank the reviewer for the comment. According to the reviewer’s suggestion, we added the following paragraph in the Discussion (Page 14, lines 354-363).

Reviewer 3 Report

This is a well developed research study based in sound evidence. Although the methodology was quite involved, there was sufficient detailed information to allow reader understanding.

There were a few suggested minor editorial edits and need for some clarification for recording methods used for the nystagmus.

Overall, very good, high resolution used for stained tissue samples presented. In addition, diagrams and graphed information were well presented and easy to understand. Statistical software however needed to be updated. Please see comments in attached documents. 

Overall, this work presents important and clinically relevant work.  

Author Response

We thank the reviewer for the comments. The editorial problems which the reviewer pointed out were rewritten. For statistical analysis, we renewed the software last year and we performed the analysis using new one again. So the statistical analysis section was rewritten (Page 5, Lines 202).

I’m so sorry for unclear describing of the method of observing nystagmus. According to the reviewer’s suggestion, we described the method more precisely (Page 3, Lines 125-128).